# A TLR4 Agonist Induces Osteosarcoma Regression by Inducing an Antitumor Immune Response and Reprogramming M2 Macrophages to M1 Macrophages

**DOI:** 10.3390/cancers15184635

**Published:** 2023-09-19

**Authors:** Iseulys Richert, Paul Berchard, Lhorra Abbes, Alexey Novikov, Kamel Chettab, Alexandra Vandermoeten, Charles Dumontet, Marie Karanian, Jerome Kerzerho, Martine Caroff, Jean-Yves Blay, Aurélie Dutour

**Affiliations:** 1Cell Death and Pediatric Cancers Team INSERM U1052, CNRS UMR 5286, Centre de Recherche en Cancérologie de Lyon, Université de Lyon, 69373 Lyon, Francepaul.berchard@lyon.unicancer.fr (P.B.); lhorra.abbes@lyon.unicancer.fr (L.A.); jean-yves.blay@lyon.unicancer.fr (J.-Y.B.); 2HEPHAISTOS-Pharma, 21 rue Jean Rostand, 91400 Orsay, France; an@hephaistos-pharma.com (A.N.); jk@hephaistos-pharma.com (J.K.); mc@hephaistos-pharma.com (M.C.); 3INSERM U1052, CNRS UMR 5286, Centre de Recherche en Cancérologie de Lyon, Université de Lyon, 69373 Lyon, France; abdelkamel.chettab@univ-lyon1.fr (K.C.); charles.dumontet2@univ-lyon1.fr (C.D.); 4Hospices Civils de Lyon, 69007 Lyon, France; 5SCAR, Rockefeller Medecine School, Université Claude Bernard Lyon 1, 69367 Lyon, France; alexandra.vandermoeten@univ-lyon1.fr; 6Department of Biopathology, Léon Bérard Center, Unicancer, 69008 Lyon, France; marie.karanian@lyon.unicancer.fr; 7Department of Medicine, Léon Bérard Center, Unicancer, 69008 Lyon, France; 8Department of Medical Oncology, Université Claude Bernard Lyon 1, 69008 Lyon, France

**Keywords:** osteosarcoma, immunotherapy TLR4 agonist, macrophages reprograming, growth inhibition

## Abstract

**Simple Summary:**

Due to the lack of progress in OsA’s treatment and survival, there is a need to develop new therapeutic approaches for this tumor. OsA is a complex tumor for which the immune microenvironment appears as a potential therapeutic option. This study explores the antitumor effect of a chemically detoxified TLR4 agonist MP-LPS, under liposomal formulation called Lipo-MP-LPS. The agent induces a significant antitumor response and inhibition of tumor growth in an immunocompetent OsA model. Lipo-MP-LPS acts by favoring the switch of M2 macrophages to M1, and promotes T-cell recruitment. The study suggests that Lipo-MP-LPS could be used alone or in combination with other therapies for refractory tumors like OsA.

**Abstract:**

Osteosarcoma (OsA) has limited treatment options and stagnant 5-year survival rates. Its immune microenvironment is characterized by a predominance of tumor-associated macrophages (TAMs), whose role in OsA progression remain unclear. Nevertheless, immunotherapies aiming to modulate macrophages activation and polarization could be of interest for OsA treatment. In this study, the antitumor effect of a liposome-encapsulated chemically detoxified lipopolysaccharide (Lipo-MP-LPS) was evaluated as a therapeutic approach for OsA. Lipo-MP-LPS is a toll-like receptor 4 (TLR4) agonist sufficiently safe and soluble to be IV administered at effective doses. Lipo-MP-LPS exhibited a significant antitumor response, with tumor regression in 50% of treated animals and delayed tumor progression in the remaining 50%. The agent inhibited tumor growth by 75%, surpassing the efficacy of other immunotherapies tested in OsA. Lipo-MP-LPS modulated OsA’s immune microenvironment by favoring the transition of M2 macrophages to M1 phenotype, creating a proinflammatory milieu and facilitating T-cell recruitment and antitumor immune response. Overall, the study demonstrates the potent antitumor effect of Lipo-MP-LPS as monotherapy in an OsA immunocompetent model. Reprogramming macrophages and altering the immune microenvironment likely contribute to the observed tumor control. These findings support the concept of immunomodulatory approaches for the treatment of highly resistant tumors like OsA.

## 1. Introduction

Osteosarcoma (OsA) is the most common primary bone tumor in terms of incidence. It affects preferentially children, adolescents and young adults and is the second leading cause of cancer-associated death. Current OsA treatments combining surgery of the primary tumor with neoadjuvant and adjuvant poly-chemotherapies has not evolved over the last four decades. Similarly, OsA prognosis has not improved. The overall five-year survival rate is 60–70% for patients with localized disease but falls below 30% for patients with unresectable or relapsed tumors, non-responder to chemotherapy, or patients with metastases at diagnosis [1,2,3,4]. To improve patient survival, there is an urgent need to rapidly develop new treatments for OsA. The complexity of OsA genome and its genomic instability have impaired the identification of molecular therapeutic targets [5,6]. Targeted therapies showed great promises in preclinical studies, but failed to meet their expectations in clinical trials [7,8,9]. All these data indicate that directly targeting OsA tumor cells may not be sufficient and that focusing on its immune environment may be key.

Several studies suggested that immunotherapies could be applied in OsA. Analysis of biopsies from patients revealed that the OsA environment presents intra-tumoral immune infiltrates, notably CD8+ T-cells, but especially tumor-associated macrophages (TAMs), which make up the major immune population in OsA [10,11,12]. Some studies suggest that an increased infiltration of M2-like TAMs is associated with OsA metastases and poor patient prognosis, while a high level of M1 macrophages is associated with good prognosis [13,14]. However, immunotherapies aiming to modulate macrophages activation and polarization, such as mifamurtide (MEPACT), or to stimulate adaptive immunity, like immune-checkpoint inhibitors (ICIs), did not demonstrate significant clinical benefits in OsA [15,16,17,18]. In early phase trials, ICI targeting CTLA-4 (ipilimumab) and PD-1 (nivolumab and pembrolizumab) have failed to demonstrate efficacy in OsA when given as monotherapy. These failures may be related to the inability of such immunotherapies to restore both anti-tumor macrophage function and effective adaptive immune responses.

Toll-like receptor 4 (TLR4) agonists, like natural lipopolysaccharides (LPSs), are potent innate immunostimulants enabling a direct stimulation and polarization of immune cells like macrophages but also an indirect activation, recruitment and maintenance of effective and long-lasting T-cell responses [19]. Several clinical observations support the view that TLR4 agonists may be a relevant therapeutic option for the treatment of OsA [20]. Since the use of bacterial immunotherapy by William Coley in the late nineteenth century to treat sarcomas [21], studies have shown a positive effect of systemic infection on the growth of OsA [22,23,24]. In a large cohort of OsA patients, a ten-year survival was observed in patients who developed an infection within one year of orthopaedic surgery (84.5% in the infected group vs. 62.3% in the non-infected group) [24]. These observations suggest a relationship between TLR4 and OsA progression as bacterial proliferation commonly activates TLR4 [25]. On a mouse model of OsA, TLR4 activation by systemic administration of LPSs suppressed the progression of OsA via stimulation of CD8+ T-cells [26]. All of these studies demonstrate the therapeutic interest to use TLR4 agonists in OsA.

LPSs have demonstrated potent anti-tumor efficacy, but intravenous injection of natural LPSs remains too toxic to reach an effective dose in patients [20,27]. To improve LPS tolerance, formulation in liposomes and alternative routes of administration were used but not found to be sufficient. Chemical modifications of LPSs are the most efficient approaches to eliminate their deleterious properties. The release of a phosphate group from *Salmonella minnesota* R595 lipid A generated “mono phosphoryl lipid A” (MPL, from GSK) and reduced toxicity of the molecule, while keeping its immuno-stimulant properties [28]. Natural and synthetic LPS derivatives were developed and confirmed potent antitumor activities [29,30]. Intratumoral administration of GLA-SE/G100 showed anti-tumor immune responses and tumor regressions in Merkel cell carcinoma [31]. Although intratumoral injection of TLR4 agonists has shown promising results, responses outside of the injected tumor sites remained limited, impairing their efficacy and capacity to address hard-to-reach cancers such as OsA [32]. Systemic administration could overcome these limitations but current TLR4 agonists are too toxic for intravenous administration, as demonstrated by recent Phase 1 clinical trials [33].

MP-LPS is an innovative chemically detoxified monophosphorylated LPS (Patent EP20305264) further formulated in liposomes to obtain Lipo-MP-LPS. Thanks to its unique structure-preserving core sugars linked to MPLA, it is the first TLR4 agonist compatible with IV administration. Preclinical studies showed that Lipo-MP-LPS presents a better safety profile than TLR4 agonists in development, while maintaining strong adjuvant effects in combination with therapeutic monoclonal antibodies [34].

In this study, after having identified in silico that high TLR4 pathway activation is associated with lower metastasis spreading in OsA, we investigated the efficacy and safety of Lipo-MP-LPS in vivo in an immunocompetent OsA model. We demonstrated that as monotherapy, Lipo-MP-LPS administration caused 50% of complete tumor regression and significant survival prolongation. Mechanistic studies suggested that Lipo-MP-LPS efficacy is associated with its capacity to reshape the tumor micro-environment (TME) through the systemic activation of innate immune cells (notably M1 macrophages) and Th1-type adaptive immune responses (CD8/CD4 T-cells), and their recruitment at tumor site.

## 2. Materials and Methods

### 2.1. Bioinformatic Analyses

Two publicly available OsA datasets (GSE21257, GSE33382, contents summarized Figure 1A) were used. Background correction and raw dataset normalization were performed by the Gilles Thomas bioinformatics platform (CRCL, Lyon, France). The GSE21257 dataset was used to analyze the correlation between OsA’s immune environment composition and tumor’s metastatic potential. This immune metastatic signature was then confirmed on the other dataset. The Microenvironment Cell Populations (MCPs)-counter algorithm was applied to determine the prognosis value of immune populations present and of the most encountered cytokines encountered in primary OsA’s microenvironment [35]. The Broad Institute Molecular Signature Database (MSigDB) v6.2 (with a R implementation of the ssGSEA algorithm (package R GSVA [36])) enabled determination of the activation score of immune signaling pathways, antigen presenting pathways and TLR pathways. These analyses were performed by the Gilles Thomas bioinformatics platform (CRCL, Lyon, France). All survival rates were estimated using the Kaplan–Meier method with 95% confidence intervals (CI) (GraphPad prism version 6.00-GraphPad software, La Jolla, CA, USA.

### 2.2. In Vivo Experimental Model and TLR4 Agonist (Lipo-MP-LPS) Formulation

Rat osteosarcoma model

For all experiments, 4 to 9 weeks old Sprague-Dawley rats (Charles River, L’Arbresle, France) were maintained in the in the pathogen-free animal facility “SCAR” (Agreement # A 69 388 10 01, Rockefeller medicine university). The transplantable orthotopic rat OsA model used has been previously described [37,38]. For all surgical procedure, pre-analgesia was induced by a subcutaneous injection of buprenorphine (0.05 mg/kg) (ECUPHAR, Belgique). Tumor implantation was performed as previously described on anesthetized animals (isoflurane/oxygen (2.5%/1.5%, *v*/*v*) (Minerve, Esternay, France) [38].

Modified TLR4 agonist (Lipo-MP-LPS)

The TLR4 agonist Lipo-MP-LPS was produced and provided by HEPHAISTOS-Pharma (Orsay, France). Briefly, LPS from *Bordetella pertussis* was extracted with non-toxic solvents from a culture pellet, as previously described and patented (Patent WO2004062690A1). The extracted LPSs were then chemically modified to obtain safe molecules, by removal of the glycosidic phosphate group of the lipid A moiety, and by partial de-O-acylation, keeping most of the immunostimulatory effect of the native molecule as previously described (Patent EP20305264). Structural quality control analyses were performed on the product by matrix assisted laser desorption mass spectrometry (MALDI-MS-Shimadzu AXIMA Performance time-of-flight mass spectrometer, in linear mode with delayed extraction) and thin-layer chromatography as previously described [39]. The level of supramolecular aggregate formation, as well as solubility, both in water, were assessed by dynamic light scattering (DLS). Chemical purity, including amino and nucleic acids contents, was evaluated, respectively, by LC–MS (Hitachi L-8800 amino acids analyser with a 2620MSC-PS column) and UV spectrophotometry (absorbance at 260 nm-Denovix spectrophotometer). Immunological purity was also controlled by evaluating the level of activation of TLR4 and TLR2 pathways using in vitro assays on HEK-BlueTM TLR4/TLR2 cells (InvivoGen, San Diego, CA, USA; Diaclone, Besançon, France). This TLR-4 agonist was encapsulated in liposomes before being used in preclinical experiments. The production of liposomes and the encapsulation of TLR-4 agonist in liposomes were performed by the antibody and cancer team (CRCL-Lyon France) according to a well-established protocol [34]. In vivo studies demonstrated that MP-LPS presents a better safety profile than native LPS and competitor TLR4 agonists [34]. Rabbit pyrogen tests (RPTs) indeed showed that the chemically detoxified MP-LPS presents a pyrogenic activity 230 times reduced compared to native LPS (with a maximum non-pyrogenic dose (MNPD) of 175 ng/kg vs. 0.75 ng/kg), further 10-fold improved by the liposomal formulation.

### 2.3. Lipo-MP-LPS Efficacy Study

When tumors were established and growing, animals were treated either with saline (n = 7), Lipo (liposome, 2.5 mL/kg) (n = 8), or Lipo-MP-LPS (250 µg/kg) (n = 8). Treatments were administered intravenously twice a week over a period of 3 weeks or until tumors reached 2500 mm^3^. Tumor volume was monitored over the treatment period according to the formula: V = 0.5 × L × S2, where L and S are the largest and the smallest tumor diameters. At time of euthanasia tumors, spleen and blood were harvested. Tumors were fixed in 4% formalin for IHC analyses. Spleens were sampled in RPMI medium supplemented with heat-inactivated fetal bovine serum (FBS; 10%;) and penicillin-streptomycin (1%) (all from Life Technologies, Villebon-sur-Yvette, France). Plasma was obtained by centrifugation of collected blood (1700× *g*, 10 min, 4 °C) and stored at −80 °C for further analyses.

### 2.4. Immune Infiltrates Analysis at Early Stage of Antitumor Response

Early changes in tumor immune infiltrates (cellular composition and cytokines released) induced by Lipo-MP-LPS were monitored at the onset of an anti-tumor response (i.e., when an arrest of tumor progression was observed). For this purpose, OsA bearing rats were treated, as previously indicated, with either Lipo-MP-LPS (n = 5), saline (n = 3) or liposomal Lipo (n = 3) till an anti-tumor response was observed (i.e., after 3 IV injections). At the time of euthanasia, tumors were sampled for analysis of immune cells infiltrates (flow cytometry), and of cytokines released.

### 2.5. Flow Cytometry

Single cell suspension preparation and staining

Tumor immune infiltrates and splenocytes were isolated and analyzed by flow cytometry. To isolate intratumoral immune infiltrate, tumors were cut into 2 mm^3^ pieces before being enzymatically digested (MACS Tissue Dissociation Kits; 37 °C, 40 min) according to manufacturer’s instructions (Miltenyi Biotech, Gladbach, Germany). After filtration (70 μm cell strainer (Miltenyi Biotech), centrifugation (150× *g*, 5 min) and removal of red blood cells (Red Blood Cell Lysis Solution (Miltenyi Biotech)) cells’ suspension was washed (PBS) and resuspended in staining buffer (3% BSA in PBS) before being incubated with antibody panels enabling the phenotyping of main immune populations (Appendix A).

For splenocytes isolation, spleens were cut into 2 mm^3^ pieces before being mechanically dissociated. After filtration (30 µm cell strainer (Miltenyi Biotech), centrifugation (150× *g*, 5 min) and incubation with Red Blood Cell Lysis Solution (2 min), splenocytes were aliquoted in frozen media (50% DMEM, 40% FBS and 10% DMSO) and stored in liquid nitrogen for further analyses. For flow cytometry analyses, splenocytes were thawed, resuspended in staining buffer (3% BSA in PBS) before being incubated with antibodies panels (Appendix A).

Intracellular CD68 staining was performed after initial CD45, CD163 and CD86 staining (dilution 1/10 in staining buffer, final volume 50 µL, 10 min, 4 °C) fixation and permeabilization in the FOXP3 staining buffer set (Miltenyi Biotech).


*FACS analyses*


All samples were resuspended in PBS before acquisition on a flow cytometer BD LSRII flow cytometer (BD Biosciences, Franklin Lakes, NJ, USA) data were analyzed using Flowing Software (version 2.5.1; BD Biosciences). Gating strategies used to identify the different immune populations from the spleen and tumors infiltrates are presented in Appendix A.

Intratumoral immune infiltrates are presented as the percentage of CD45+ cells. The percentage of each immune population among CD45+ splenocytes was normalized using the ratio spleen’s weight/animal’s weight. For each treatment, results are presented as the fold change of each immune effector, relative to control group, with an arbitrary set up at 1.

### 2.6. ELISpot Analysis

The Rat IFN-γ ELISpot kit (R&D, Minneapolis, MN, USA) was used to measure IFN-γ released by splenocytes of 3 animals from the different treatment groups. Splenocytes were thawed and seeded at 40,000 cells/well into ELISpot plates (R&D Systems) at 37 °C for 24 h. After a thorough wash, biotinylated secondary antibody was added, according to manufacturer’s recommendations. Spots were revealed and counted with a Series 1 ImmunoSpot Image Analyzer and software (Cellular Technologies, Cleveland, OH, USA). Splenocytes from healthy rat stimulated with PMA (5 ng/mL) and ionomycin (500 ng/mL) (Sigma–Aldrich, St Quentin Fallavier, France) for 24 h were used as positive control.

### 2.7. Bone Marrow Derived Macrophage (BMDM) Generation and Differentiation toward M1 and M2 Phenotype

BMDMs were isolated from the marrow of the femurs from 8- to 10-week-old Sprague-Dawley rats according to the following method adapted from [40]. Briefly, after removing skin and muscle tissues from the bones, bones were sprayed with 70% ethanol and transferred to sterile hood and cut at both ends. Marrow was flushed in RPMI complete medium, cell suspension was filtered through a 70 μm cell strainer (Miltenyi Biotech) and centrifuged (150× *g*, 5 min). Supernatant was removed, the pellet resuspended in red blood cell lysis buffer (Miltenyi Biotech) and centrifuged (150× *g*, 5 min). The cells pellet was then washed in RMPI complete medium, centrifuged (150× *g*, 5 min) and resuspended in RPMI Complete medium. Cells were seeded in sterile 10-cm Petri dishes, and non-adherent cells were removed from the dish after 2 h by thorough washing with RPMI medium. The method to generate macrophages was adapted from [40,41,42]. Isolated monocytes were resuspended in complete medium (DMEM medium supplemented by 10% heat inactivated FBS; 1% penicillin streptomycin, all from Life Technologies) and seeded at 100,000 cells/cm^2^ in 10-cm Petri dishes in complete medium supplemented with M-CSF (20 ng/mL; Biolegend, Montigny-le-Bretonneux, France) for 7 days to induce their differentiation in M0 macrophages. M0 macrophages were incubated with complete medium supplemented either by IFN-γ (50 ng/mL; R&D System, Abingdon, UK) or by TGF-β (50 ng/mL; Peprotech, Neuilly-sur-Seine, France) for 24 h to be, respectively, polarized toward M1 immunoactivating or M2 immunosuppressive macrophages. These M1- or M2-oriented macrophages were then treated with either Lipo (5 µL/mL) or Lipo-MP-LPS (50 ng/mL) for 24 h. IFN-γ and TGF-β were, respectively, added to M1 or M2 macrophages to maintain their acquired phenotype and serve as control. After 24 h of culture, supernatants were collected to measure the cytokines released, while macrophages were harvested and phenotyped by flow cytometry.

### 2.8. Functional Analyses of Lipo-MP-LPS Treated Macrophages

Phagocytosis assay

The rat osteosarcoma cell line UMR106 (ATCC, Manassas, VA, USA) was incubated for 24 h with Crizotinib 10µM; (Selleckchem, Souffelweyersheim, France) to induce apoptosis. Apoptotic cells were harvested, washed in PBS and stained with CFSE (500 nM; 30 min at 37 °C). After being washed in complete medium, apoptotic UMR106 cells were co-cultured with previously established M1, M2 macrophages pretreated or not by Lipo-MP-LPS at a Target/Effector ratio 5:1 at 37 °C. To exclude false positives, similar cocultures were incubated at 4 °C. After 2 h of incubation adherent cells were collected and labelled with CD45 antibody and analyzed by FACS on the LSR Fortessa cytometer (BD Biosciences). Data were analyzed using Flowing Software. The percentage of CD45+/CFSE+ cells was considered the percentage of macrophages having phagocytosed tumor cells.

Cytokines profile assay

The rat inflammation Legendplex^TM^ panel, (Biolegend, San Diego, CA, USA) was used to measure simultaneously the level of 13 secreted cytokines in the supernatants of BMDM and tumor extracts. Experiment was performed following the manufacturer’s instructions. Cytokines were measured in undiluted BMDM supernatants, and 1/4 diluted tumor protein extract using the LSR Fortessa (BD Biosciences) and analyses were performed using the Legendplex software (Biolegend). Cytokines that were below the detection limit were considered null (i.e., CCL2; CXCL1; GM-CSF; IFN-γ; IL-1α; IL-1; IL-6; IL-10; IL-12(p70); IL-17A; IL18; IL-33;TNF-α). Plasmatic IFN-γ was measured in the plasma of controls and treated animals with the Rat IFN-γ ELISA Kit (ELISA Genie) according to manufacturer instructions. Samples’ absorbance was read on a TECAN’s Infinite^®^ F500 at 450 nm.

### 2.9. Immunohistochemistry

All immunohistochemistry (IHC) analyses were performed on 5 μm FFPE sections. Ki67, CD3 and CD163 staining were performed on an automated Ventana Discovery XT staining system (Ventana Medical Systems, Tucson, AZ, USA) according to the manufacturer’s instructions.

CD68, CD8, TLR4 immunostaining were performed manually. After antigen retrieval (sodium citrate buffer pH 6, 30 min, 95 °C), sections were incubated overnight at 4 °C with either anti-CD8, anti-CD68 or anti-TLR4 primary antibody. Details on antibodies, clones, manufacturers and staining conditions for IHC are listed in Supplementary Methods, Appendix A. After a washing step, endogenous peroxidases were inactivated (H_2_0_2_ 0.3%, 15 min, RT). The primary antibody was detected using biotinylated goat anti-mouse secondary antibody (dilution 1:100, 1 h, RT) followed by avidin-biotin complex and DAB peroxidase (dilution 1:300, 30 min, RT). Sections were counterstained with hematoxylin. All reagents are from Vector Lab (Burlingame, CA, USA). All slides were examined under a Zeiss Axioimager. Histological changes, necrosis rates and Ki67+ cell counts induced in response to Lipo-MP-LPS were analyzed by a pathologist (MK) according to the criteria used to assess clinical OsA response rates. Density of each immune cells in tumor infiltrate was appreciated by counting CD3, CD8 or CD163+ cells in five non-contiguous representative areas with the highest density of infiltrates cells using ImageJ software (https://imagej.net/, accessed on 12 July 2019 and 8 July 2023, University of Wisconsin-Madison).

### 2.10. Statistical Analyses

Data are expressed as mean ± SD.

For bioinformatics assays, in the GSE21257 dataset a Wilcoxon test was applied to analyze the potential correlation between expression of immune markers and OsA 5Y free-metastasis survival.

For in vivo experiments, one representative out of three similar Lipo-MP-LPS efficacy assays is presented. The non-parametric Mann–Whitney test was applied to determine whether one treatment differed significantly from all others.

For in vitro experiments, Student’s *t* test was run considering a normal distribution of samples and the correction of Welch was applied due to the variability of SD between groups. Statistical significance was determined at the alpha = 0.05 level. Statistical analyses were performed using R Studio or GraphPad Prism software (version 6, GraphPad).

## 3. Results

### 3.1. A transcriptomic Pro-Inflammatory Immune Profile Is Associated with Delayed Development of Metastases in OsA

The potential prognosis value of immune markers in OsA was evaluated on the GSE21257 dataset regrouping 53 OsA biopsies at diagnosis (Figure 1A). MCP analyses showed that patients who did not develop metastases in the 5 years after diagnosis had a higher expression of genes related to cytotoxic T-cells, NK cells, cells of monocytic lineage and neutrophils in their primary OsA microenvironment (Figure 1B). A higher expression of pro-inflammatory cytokines such as IL-1β, IL-6 and IFN-γ was also observed (Figure 1C). The MSigDB signature indicated an increased expression of genes related to antigen presenting and TLR signalling pathways among which TLR4, MyD88 and HMGB1 [43] (Figure 1D). Interestingly, a higher expression of TLR4 was associated with better metastasis free survival (*p* = 0.01, HR:0.39, CI: 0.19–0.8) (Figure 1E).

These analyses were confirmed on the GSE33382 dataset and seem to indicate that a pro-inflammatory environment and the activation of TLR pathways prevent metastasis spreading in OsA. Based on these observations and as macrophages are the most abundant immune effectors present in OsA microenvironment, an immunomodulatory approach using TLR4 agonist to reorient macrophages towards a pro-inflammatory phenotype could be considered.

### 3.2. Lipo-MP-LPS Induced Regression of Primary OsA

The efficacy of Lipo-MP-LPS as single agent in IV injection was evaluated in an OsA tumor model in immunocompetent rats. Lipo-MP-LPS was found to significantly inhibit OsA progession compared to control and vehicule groups (Figure 2A). At day 14 of treatment (4th injection), tumor growth was inhibited by 74.9 ± 47.3% in Lipo-MP-LPS (*p* < 0.005 vs. cte; *p* < 0.05 vs. vehicle) (Figure 2B). Lipo-MP-LPS was also able to induce complete responses (CR) in 50% of the animals (vs. 0% of CR for control and vehicle groups), and stabilized responses (SRs) in the other 50% were defined by slowed tumor growth (Figure 2C). This inhibition of tumor progression was confirmed by a drastic decrease of the tumor cell proliferation, as seen on histological stainings. Indeed, histological analyses of control tumors showed typical OSA structures, i.e., the presence of numerous hyperproliferative areas was characterized by a high cell density and an elevated mitotic score (KI67 positive cells >90%; Figure 2D). In comparison, tumors from Lipo-MP-LPS treated group presented histological rearrangements. At the initial tumor site of complete responders (CRs), the presence of tumor remnant was observed and composed mainly of necrotic areas, fibrosis and less than 20% KI67 positive cells. In SR tumors, an increase in necrotic areas was observed, which is associated with a decrease in proliferative area and a decrease in Ki67 positive cells (50–60%; Figure 2D).

Organotypic culture assay showed that Lipo-MP-LPS efficacy in our model was not due to direct effect of this product on OsA cells (Appendix A).

Interestingly, while all animals in the control group showed a similar tumor growth profile and reached endpoints at day 18, we observed that 50% of the animals in the vehicle group presented tumor growth stabilization (Figure 2C). We first attributed this effect to a non-specific stimulation of the immune system as some liposomes are known to be eliminated by phagocytic cells such as macrophages [44]. However, subsequent analyses showed that this was in fact due to a cross contamination of the vehicles by a small amount of MP-LPS during their production. LC-MS2 analyses conducted at the end of our investigations indeed revealed those vehicles were not empty but contained MP-LPS corresponding to a dose of 1.4 µg/kg MP-LPS at each injection. Moreover, additional preclinical studies confirmed that similar empty liposomes (Not containing MP-LPS as controlled by LC-MS2) have no anti-tumor efficacy in several mouse tumor models, and no immunostimulatory effects on human and murine macrophages [31]. However, we were not able to confirm this in our OsA model due to the discontinuation by all suppliers of the rat strain (Sprague-Dawley OFA) used in our studies. All our studies were therefore carried out with 2 doses (250 µg/kg and 1.4 µg/kg) of MP-LPS formulated in liposomes (Lipo-MP-LPS High and Low), enabling us to make a first exploration of the relationship between Lipo-MP-LPS dose and its impact on OsA tumors.

### 3.3. Lipo-MP-LPS Changed the Composition of OsA Immune Microenvironment

The impact of Lipo-MP-LPS treatment on immune cell populations (CD8+ TILs, CD68+ TAMs and CD163+ M2 TAMs) and TLR4+ cells density in OsA tumor microenvironments were evaluated in each tumor and linked to the individual response to treatment: non response (NR), stabilized response (SR) or complete response (CR). Similar density of CD8+ T-cells was encountered in the tumors from control, NR Lipo-MP-LPS Low and CR Lipo-MP-LPS High group (mean density: 37 ± 20 CD8+ T-cells/Field Of View (FOV) (Figure 3). However, a significantly higher density of CD8+ T-cells was observed in tumors from SR high-dose Lipo-MP-LPS group compared with the control group (*p* < 0.005) (Figure 3).

Higher levels of TLR4+ cells and CD68+ TAMs, but not CD163+ M2 TAMs, were also observed in SR and CR tumors from Lipo-MP-LPS High group compared to control and Lipo-MP-LPS Low groups (Figure 4). These data suggest that Lipo-MP-LPS efficacy is associated with the recruitment of CD8+ T-cells and M1 macrophages in OSa tumors, and that this effect is dose-dependent.

As changes in tumor immune populations were mainly observed in stabilized responders (SR), i.e., when a tumor started to regress or stopped its growth, we analyzed more in depth the tumor immune infiltrate at this stage of tumor response (Figure 5).

At the time of tumor response initiation (i.e., post 3 IV injection), a higher level of tumor infiltration by CD4 T-cells was observed in the high-dose Lipo-MP-LPS group, but not in the low-dose group, compared to control group (*p* < 0.005) (Figure 5A). IHC staining showed that those T-cells were not Treg since the proportion of FOXP3+ cells were similar between all treatment groups (12 FOXP3+/FOV in the controls groups versus 11 FOXP3+/FOV in the Lipo-MP-LPS High group). No significant difference in the density of CD8+ T-cells, NK cells and total CD68+ macrophages was observed between the different groups but higher levels of CD68+CD86+ M1 macrophages were observed in the high- and low-dose Lipo-MP-LPS groups. This suggests that Lipo-MP-LPS does not induce an early recruitment of M1 macrophages in tumors, but that it can polarize tumor-associated macrophages toward an M1 phenotype, even at very low doses. Analysis of the cytokine profile in tumors also showed that Lipo-MP-LPS treatment (high and low doses) induces a moderate proinflammatory cytokine environment in tumors characterized by increased levels of some proinflammatory cytokines (IL1*α*, IL1*β*, IL-6, IL-10, and GM-CSF) and chemokines (CCL2/MCP1 and CXCL1) compared to control group (Figure 5B). These data demonstrate that high-dose Lipo-MP-LPS can induce both an early recruitment of CD4+ T helpers at the tumor site and the polarization of TAM toward an M1 phenotype, which could help enhance a pre-existing antitumor immune response. An important number of giant multinucleated cells (GMCs) were also observed in the tumors from high-dose Lipo-MP-LPS group but not in control and low-dose Lipo-MP-LPS groups. The morphology of these cells is consistent with oteosclasts suggesting that Lipo-MP-LPS increases the number of osteoclasts and that this effect is dose-dependent (Appendix A).

### 3.4. Lipo-MP-LPS Induced a Systemic Inflammatory Response

As we showed that the IV injection of Lipo-MP-LPS increases the infiltration of immune cells in tumors, the impact of this product on immune cell populations in periphery was also analyzed. In the spleen, higher levels of CD8+ T-cells (2.83-fold; *p* < 0.005), CD4+ T-cells (2.23-fold; *p* < 0.005) and B cells (1.7-fold increase; *p* < 0.0005) were observed in the high-dose Lipo-MP-LPS group compared to control and low-dose Lipo-MP-LPS groups (Figure 6A). This was associated with a significant increase of spleen cellularity (1.91-fold; *p* < 0.05) (Figure 6B) and a higher production of IFN-γ by splenocytes (Lipo-MP-LPS High:311 ± 284 spots; Control: 54 ± 16 spots; Lipo-MP-LPS High: 116 ± 43 spots) (Figure 6C). Higher plasma IFN-γ levels were also detected in the high-dose Lipo-MP-LPS group compared to other groups (Lipo-MP-LPSHigh: 41.2 ± 10.7 pg/mL; Control: 28.4 ± 3.0 pg/L; Lipo-MP-LPSLow: 30.6 pg/mL ± 1.9) (Figure 6D). These data show that Lipo-MP-LPS can induce a systemic activation of adaptive immune responses with a Th1 profile and that this effect is dose-dependent.

### 3.5. Lipo-MP-LPS Directs Macrophages toward a Pro-Inflammatory Phenotype

Our in vivo analysis suggest that Lipo-MP-LPS has an impact on tumor-associated macrophage polarization, which have been associated with OsA tumor progression and metastatic spreading [45]. To confirm this, we assessed in vitro its impact on the phenotype and functions of bone morrow derived M1 and M2 macrophages. M1 macrophage were characterized by the high expression of CD86 and the high level of production of pro-inflammatory cytokines (TNF-α, IL-18, and IL-6) and chemokines (CXCL1 and CCL2). In contrast, M2 macrophages were characterized by a high expression of CD163, but not CD86 and the low level of production of pro-inflammatory cytokines and chemokines.

In vitro stimulation of M1 polarized macrophages with the high-dose Lipo-MP-LPS, but not low-dose, was shown to induce a slight decrease in CD163 expression (0.70-fold; NS) but a more than 2-fold increase in CD86 expression (2.8-fold ± 1.6; NS) (Figure 7A). M1 macrophages were also found to secrete higher levels of proinflammatory cytokines and chemokines, such as TNFα, IL18, IL6 and CXCL1, when stimulated with high-dose Lipo-MP-LPS than with PBS, IFNγ or low doses of Lipo-MP-LPS (Figure 7B). These data demonstrate that high-dose Lipo-MP-LPS maintained M1 pro-inflammatory phenotype. In vitro stimulation of M2 polarized macrophages with high-dose Lipo-MP-LPS, but not low-dose, was shown to significantly increase the expression of CD86 at high-dose (6.6-fold; *p* = 0.01) but not to alter the expression of CD163 (Figure 7A). M2 macrophages were also found to secrete higher levels of proinflammatory cytokines and chemokines such as TNFα, IL6 and CCL2 (like stimulated M1 macrophages) when stimulated with high-dose Lipo-MP-LPS than with PBS, IFNγ or low doses of Lipo-MP-LPS (Figure 7B). These data suggest that high-dose Lipo-MP-LPS, but not low-dose, can repolarize M2 macrophages toward a M1 phenotype. Lipo-MP-LPS stimulation were also found to decrease the phagocytic activity of M1 and M2 macrophages (Figure 7C).

## 4. Discussion

The treatment and five-year survival rates of OsA have remained unchanged for decades. This lack of progress highlights the need of new therapeutic approaches in this indication. Over the last 20 years, the focus has been on the genetic exploration of OsA but the wide genomic complexity of it has not allowed the identification of recurrent therapeutic molecular targets. OsA arises in a very complex environment with a finely balanced interplay between different cell types, where pathways involved in bone homeostasis and immune responses are interconnected and share common effectors [44].

OsA’s immune microenvironment is very heterogeneous as it involves tumor-associated macrophages (TAMs), dendritic cells, myeloid cells, osteoclast and lymphocytes [46]. TAMs are the most abundant immune population present in OsA’s immune environment and their role in this tumor’s progression is not very clear. When focusing on the implication of primary OsA’s immune microenvironment in its metastatic dissemination, the role of macrophages is also controversial. Some studies showed a significant increase of M1 macrophages in primary tumors of non-metastatic patients. In contrast, others established that a high proportion of CD163+ M2 macrophages in primary OsA was associated with better survival and slower metastatic dissemination [14,45,47]. A recent transcriptomic OsA classification linked to biological functions confirmed that favorable prognosis tumors are associated with specific innate immune expression [14]. In concordance with this, our bioinformatics analyses showed that a proinflammatory microenvironment in the primary tumor prevents OsA’s metastatic dissemination. Our data notably indicate that TLR signaling pathways (among which TLR4) that induce the production of proinflammatory cytokines prevent the metastatic process of OsA. Moreover, our data showed that a higher expression of TLR4 was associated with better metastasis-free survival.

These elements set up the rationale of our work for evaluating the antitumor effect of a TLR4 agonist against OsA tumors. In the present study, we evaluated the efficacy of Lipo-MP-LPS, an innovative liposomal formulated chemically detoxified bacterial TLR4 agonist. Lipo-MP-LPS was previously found to present a better safety profile than current TLR4 agonists in development, while maintaining potent antitumor activity and strong adjuvant effects in combination with therapeutic monoclonal antibodies [34].

We demonstrated that Lipo-MP-LPS can induce a dramatic antitumor response in an orthotopic OSa tumor model in immunocompetent rats. Lipo-MP-LPS was indeed able to induce 50% of complete tumor regression in this OsA tumor model and to delay tumor progression in the other 50%. On average, this compound inhibited tumor growth by 75%, a level of inhibition never seen with any of the immunotherapies previously tested in OsA. Lipo-MP-LPS showed better efficacy than L-mifamurtide (MEPACT), an immunomodulator of macrophages and monocytes. In preclinical studies, MEPACT in combination with zoledronic acid only inhibited OsA tumor growth by 41% without inducing tumor regressions [48].

Surprisingly, tumor growth stabilizations were observed in 50% of the animals from the vehicle group. We first attributed this effect to a non-specific stimulation of the immune system, as some liposomes are known to be eliminated by phagocytic cells such as macrophages [49]. However, subsequent analyses showed that this effect was due to a contamination of the vehicles by a small amount of MP-LPS. LC-MS2 analyses indeed showed that our vehicles contained 1.4µg/kg of MP-LPS due to cross contaminations during their manufacturing. This small amount is sufficient to explain our observations, as TLR4 agonists are known to be potent immunostimulants able to stimulate the innate immune system at very low-dose levels of the order of a picogram [50]. This allowed us to explore the relationship between Lipo-MP-LPS dose and its impact on OsA tumors. On this basis, our data showed that a very low dose of Lipo-MP-LPS is sufficient to impact OsA tumor growth but that a higher dose is required to induce complete tumor regressions. This demonstrates a dose effect of Lipo-MP-LPS on the progression of OsA tumors.

The absence of expression of TLR4 on tumor cells excluded a direct cytotoxic effect of Lipo-MP-LPS as confirmed by the absence of caspase-3 mediated cell death in organotypic model of OsA treated by Lipo-MP-LPS (Appendix A; Lipo treated OsA: 205 Cleaved Caps 3+ Cells/FOV, Lipo-MP-LPS treated OsA: 189 Cleaved Caps 3+ Cells/FOV). We showed that the antitumor effect of Lipo-MP-LPS involved the stimulation of the immune system and the immune modulation of OsA’s environment.

At early stages of tumor response (i.e., when tumor’s progression stopped), we indeed showed that Lipo-MP-LPS induces an early tumor infiltration by CD3+CD4+CD25+ T helper cells, but not Treg, and increases the levels of CD68+CD86+ M1 macrophages in tumors. This was associated with the promotion of a moderate proinflammatory cytokine environment in tumors characterized by increased levels of proinflammatory cytokines (IL1α, IL1β, IL-6, IL-10 and GM-CSF) and chemokines (CCL2/MCP1, and CXCL1). Interestingly, we showed that Lipo-MP-LPS increased the levels of M1 macrophages in tumors without modifying the proportion of total macrophages. This suggests that at this stage, Lipo-MP-LPS does not induce the recruitment of M1 macrophages but induces a polarization of TAMs toward an M1 phenotype. This was confirmed by in vitro studies on M1- and M2-polarized BMDM. We indeed showed that Lipo-MP-LPS can maintain in vitro the pro-inflammatory phenotype of rat M1 macrophages but also turn M2 macrophages toward a M1 phenotype. These data are consistent with previous observations made with Lipo-MP-LPS on human M1-M2 macrophages [34]. This M2-M1 phenotype switch caused by Lipo-MP-LPS was also reported with other immunostimulants, including TLR4 agonists but also the MEPACT [34,51,52]. Our data thus showed that Lipo-MP-LPS can induce an early reshape of the OsA microenvironment through the recruitment of CD4+ T-cells and the polarization of TAMs toward an M1 phenotypes. The early local activation and polarization of TAM also suggest that Lipo-MP-LPS can reach the tumor, as confirmed by biodistribution studies [34].

Early tumor infiltration by activated CD4+ T-cells induced by Lipo-MP-LPS is not due to a direct effect, since lymphocytes do not express TLR4 [34]. Our hypothesis is that CD4 T-cell activation is mediated by the release of tumor antigens by Lipo-MP-LPS-polarized M1 macrophages, which are known to have anti-tumor effects [53], and that their recruitment into the tumors is mediated by the chemokines produced in the tumor microenvironment. However, some of our data suggest that these two observations are not related. We indeed observed that a low dose of Lipo-MP-LPS did not induce tumor infiltration by CD4+ T-cells, although it increased the levels of M1 macrophages in tumors and production of proinflammatory cytokines and chemokines. M1 macrophages stimulated with low-dose Lipo-MP-LPS may however present reduced anti-tumor effects as some studies showed that low doses of TLR4 agonists may turn macrophages toward a tolerant state and M2 phenotype [54]. Our in vitro investigations on M1 and M2 polarized macrophages support this hypothesis. We indeed observed that high-dose Lipo-MP-LPS, but not low-dose, can repolarize M2 macrophages toward a M1 phenotype and maintain M1 pro-inflammatory phenotype.

In osteosarcoma, the modulation of the M2/M1 ratio observed with Lipo-MP-LPS should be taken with cautions for two reasons. First, while macrophages are classified as M1 or M2 phenotype, these states represent two ends of a wide spectrum of contrasting functions. It was recently shown that M1 and M2 signatures are not mutually exclusive and that M1 and M2 signatures may coexist, which is something we may have to take into account in the near future [55]. Recently, a double CD68+/CD163+ positive TAM population with unknown polarization was identified in OsA [56]. Second, CD68+/CD163-staining is a way to identify osteoclast. An effect of Lipo-MP-LPS on osteoclast and on the balance between OCs and macrophages thus should not be excluded, particularly because numerous studies validated the ability of LPS to transdifferentiate M1 macrophages into osteoclasts [57,58,59]. Such a phenomenon could happen in the Lipo-MP-LPS-treated tumors and explain the elevated numbers of GMC (often considered in the bone to be osteoclast) observed in tumors after Lipo-MP-LPS treatment. The presence and proportion of osteoclasts present in response to Lipo-MP-LPS should be further explored as osteoclasts and macrophages are interconnected population and share a common progenitor [60].

In addition to the early modification of the tumor microenvironment, our data showed that at later stages of tumor response, high-dose Lipo-MP-LPS, but not low-dose, induces a massive recruitment of M1 macrophages and activated CD8 T-cells in tumors. Both M1 macrophages and CD8 T-cells are thought to play a major role in Lipo-MP-LPS efficacy. Several studies have indeed shown that high rates of CD8 T-cells and M1 macrophages infiltrations in OsA environments are associated with better outcome and lower metastatic dissemination, suggesting that both immune cells play a dominant role in delaying OsA progression [14,61]. In other immunocompetent tumor models, similar increase in T-cells was also associated with better antitumor response [62,63,64]. These data are also consistent with the study of Yahiro et al., who reported in a murine osteosarcoma model that TLR4 activation by systemic administration of lipopolysaccharide (LPS) can suppress the progression of OsA via stimulation and infiltration of CD8+ T-cells [26].

Our immunoanalysis also suggested that the tumor infiltration by CD4 and CD8 T-cells induced by Lipo-MP-LPS is associated with the activation of CD4 and CD8 T-cell responses in periphery. Lipo-MP-LPS was indeed found to increase the proportion of CD4+ and CD8+ T-cells in the spleen. Similar results were observed in a murine OsA model in which anti-PDL1 increased the proportion of lymphoid cells [65]. Lipo-MP-LPS was also found to increase the production of IFN-γ by splenocytes and to increase the plasmatic levels of IFN-γ. This suggests that Lipo-MP-LPS can elicit Th1 T-cell immunity in vivo as it was already described with other TLR4 agonist in the vaccine field [66]. A similar effect was encountered in response to different immunotherapies that increased the number of IFN-γ secreting cells. In these studies, increased activity of splenocytes was correlated with enhanced antitumor immune response [67,68]. All these data can indicate that the systemic inflammation seen in response to Lipo-MP-LPS in our study reflects the stimulation of adaptative immunity.

As previously mentioned, the activation of CD8 T-cell responses by Lipo-MP-LPS is assumed to be indirect, as TLR4 agonists cannot bind to lymphocytes since they do not express TLR4 [34]. Our hypothesis is that the early remodelling of the OsA microenvironment induced by Lipo-MP-LPS, in particular the increased proportion of M1 macrophages, induces an early phase of tumor elimination and thus the release of tumor antigens. M1 macrophages are indeed known to have anti-tumor effects. Tumor antigens can be captured and processed by dendritic cells, which can be stimulated by Lipo-MP-LPS, as described for other TLR4 agonists [66], leading to the secondary activation of CD4 and CD8 T-cells in periphery. Data obtained with low-dose Lipo-MP-LPS support this mechanism. We indeed observed that low-dose Lipo-MP-LPS was not able to induce the activation of T-cell responses in periphery and thus tumor infiltration by CD4/CD8 T-cells, which can be related to its lack of efficacy to fully differentiate TAMs toward an M1 phenotype.

Our data suggest that the high efficacy Lipo-MP-LPS in our OsA model is associated with its capacity to reshape the tumor microenvironment through the polarization of TAMs toward an M1 phenotype, and the activation and recruitment in tumors of M1 macrophages and CD8 T cells. This mechanism of action is in line with showed that both M1 macrophages and CD8 T-cells are associated with the progression of OsA and related metastases [14,45]. These studies suggest that an immunotherapy like Lipo-MP-LPS that could increase the infiltration of both M1-like TAMs and CD8^+^ T-cells, while limiting Tregs, could be a relevant therapeutic approach for the treatment of patients with localized or metastatic OsA. The unique capacity of Lipo-MP-LPS to act on both innate and adaptive immune system (M1 macrophages and CD8 T-cell populations) could explain its better efficacy than MEPACT and ICIs in our OsA model. Indeed, while ICIs are mainly known to act on CD8 T-cells but not on macrophage populations, MEPACT is only described as a potent stimulator of monocytes and macrophages [69]. Some studies suggest that MEPACT may impact OsA tumors through the polarization of TAM toward an intermediate M1/M2 phenotype and the activation of T-cell, but it does not induce a potent activation and recruitment of CD8 T-cells into the tumors like Lipo-MP-LPS [70]. Other TLR agonists are used. Recently, the intra-tumoral injection of a TLR9 agonist was shown to induce an anti-tumoral immune response in an immunocompetent model of OsA. Similarly, as Lipo-MP-LPS, this TLR9 agonist did not change the count of total macrophages. It modulated their phenotype, causing a depletion of CD206+ M2-like macrophages [71]. The mechanism of action of Lipo-MP-LPS is a bit different as we and Chettab et al. [34] showed that Lipo-MP-LPS polarized M2-macrophages toward a M1 phenotype. A study by Cascini et al. and our study clearly highlight the therapeutic potential of macrophages phenotypes through TLR pathways for OsA treatment [71]. The fact that TLR4 can activate two signaling pathways, Myd88 and TRIF, while other TLRs only impact Myd88, is in favor of a greater efficacy of TLR4 agonists compared to others.

The important result standing out from our study is the strong antitumor effect of IV-administered Lipo-MP-LPS as monotherapy in an OsA immunocompetent model. This indicates that immunomodulatory approaches can successfully be applied to these highly resistant tumors. Another recent study using pexitartinib in xenograft osteosarcoma model reports similar results and supports this approach of reprograming macrophages for OsA treatment [72]. We propose that the improved tumor control seen is a result of Lipo-MP-LPS ’s induced T-cell activation and tumor infiltration. Dysregulation of the bone niche observed in response to Lipo-MP-LPS and characterized by a decreased of M2 macrophages could make OsA’s microenvironment more permissive to T-cell activation and recruitment. Consistent with these findings, the antitumor immune response observed was correlated with a higher activity of T-cells present in secondary lymphoid organs of Lipo-MP-LPS treated rats.

## 5. Conclusions

While response of osteosarcoma to immunotherapies has not been as good as expected, here we report the dramatic antitumor effect of an IV-administered TLR4 agonist to inhibit OsA’s progression. We provide evidence that treatment with Lipo-MP-LPS induces this drastic effect, thanks to its capacity to reshape the OsA microenvironment through activation/stimulation and recruitment of M1 macrophages and CD8 T-cells and the alteration of TAMs phenotype. Lipo-MP-LPS presumably reprogramed TAMs toward an antitumor phenotype that could participate in direct tumor cell killing or recruit additional cells. Our data provide the first proof-of-concept that the systemic activation of the innate immune system can be safely achieved and be transferred to humans for the treatment of cancers. Recent studies have also demonstrated the anti-tumor efficacy of Lipo-MP-LPS in combination with monoclonal antibodies, paving the way for the possible application of Lipo-MP-LPS in the treatment of refractory tumors such as osteosarcoma.

## Figures and Tables

**Figure 1 cancers-15-04635-f001:**
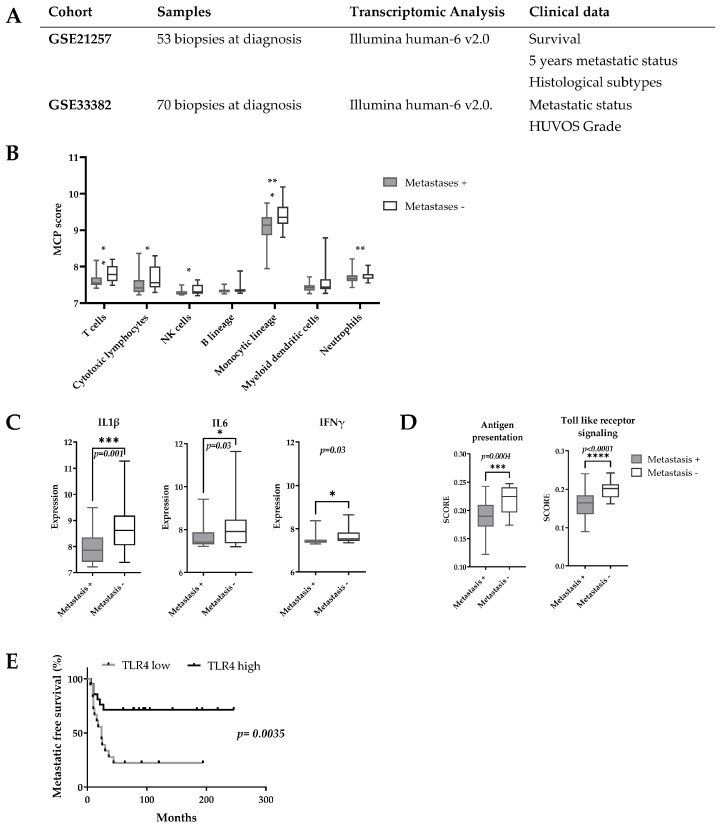
Transcriptomic pro-inflammatory signature is a prognostic marker for OsA’s metastatic spreading. (**A**) Characteristics of OsA cohorts; (**B**) MCPs score according to patients 5-years metastatic status. A higher score of genes linked to cytotoxic T-cells, monocyte lineage and NK cells was found in the 5-years metastatic-free group; (**C**) Expression of proinflammatory cytokines according to 5-years metastatic status. The expression of IL1β, IL6 and IFNγ was higher in the 5-years metastasis-free group; (**D**) Expression of antigen presentation related genes and TLR genes according to 5 years metastatic status. Higher score of activation of antigen presenting pathway, TLR pathway were found in the 5-year metastasis-free group; (**E**) Prognostic value of TLR4 expression for OsA’s metastatic spreading. A higher expression of TLR4 was associated with a significantly better metastasis free survival (*p* = 0.0035). * *p* < 0.05; ** *p* < 0.01; *** *p* < 0.001; **** *p* < 0.0001.

**Figure 2 cancers-15-04635-f002:**
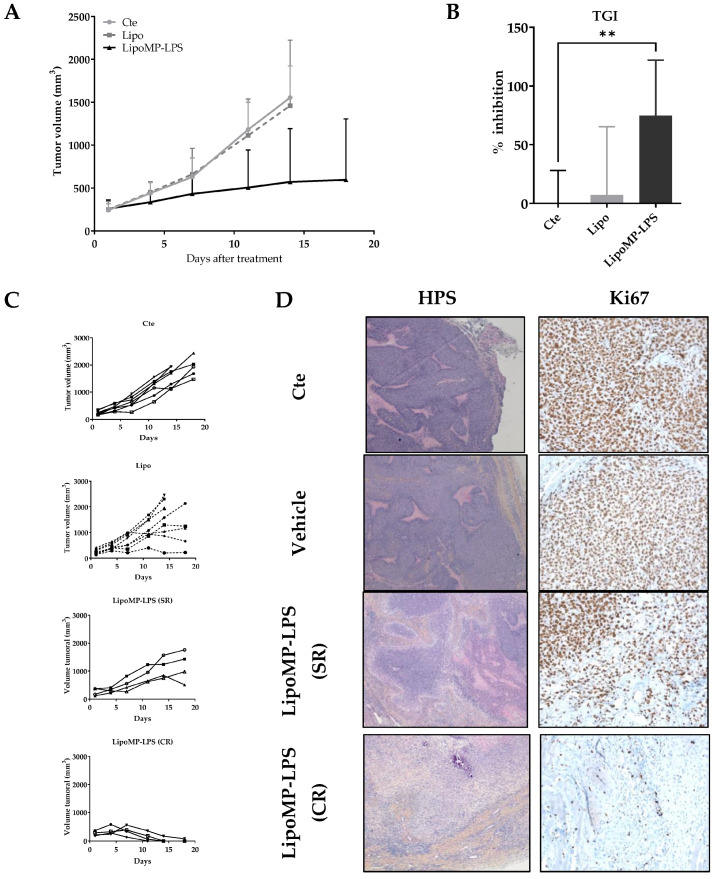
LipoMP-LPS induced tumor regression in a rat OsA model. (**A**) Mean tumor evolution of the treatments’s groups: LipoMP-LPS slowed down tumor progression; (**B**) Tumor inhibition rate at day 14: LipoMP-LPS significantly inhibited tumor growth by 75% (*p* < 0.005); (**C**) Individual tumor progression according to treatment. Control group had a homogeneous tumor progression, vehicle induced 4 stabilization of tumor progression (stabilized response SR); LipoMP-LPS induced 4 SRs and 4 complete responses (CRs); (**D**) Histological analyses (HPS and Ki67 stainings) of tumors according to treatments. HPS staining (×10) showed histological changes caused by LipoMP-LPS in the stabilized responder (SR) tumors (decrease of proliferative area, increase of necrotic areas, fibrosis) and fibrosis induced in the complete responder (CR) tumors. Ki67 staining (×40) showed a decrease in proliferative tumor cells caused by LipoMP-LPS in the stabilized tumors (SR). ** *p* < 0.005 Cte: control group (PBS), SR: stabilized response, CR: complete response.

**Figure 3 cancers-15-04635-f003:**
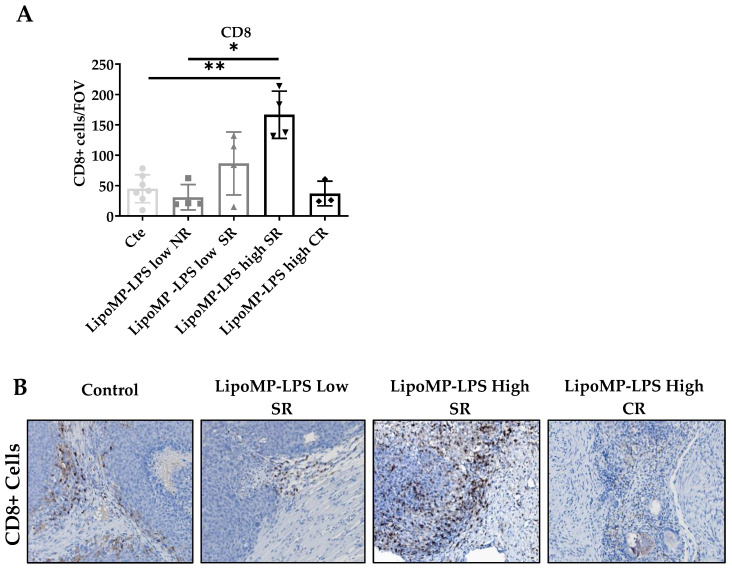
LipoMP-LPS antitumoral response was associated with increased CD8+ T-cells peritumoral cells. (**A**) Mean number CD8+ Tcells/FOV according to treatments; (**B**) Representative CD8+ staining fr each group of treatment. The mean density of CD8+ T-cells/FOV varies with tumors’s response to treatments (SR: stabilized response; CR: complete response). Magnification ×200. The induction of tumor response (i.e., tumor growth arrest (SR)) is associated with an increase in CD8+ T-cells, while the density of CD8+ T-cells in tumors that regressed (CR) is similar to the one of the control tumors. * *p* < 0.05; ** *p* < 0.005. FOV: field of view.

**Figure 4 cancers-15-04635-f004:**
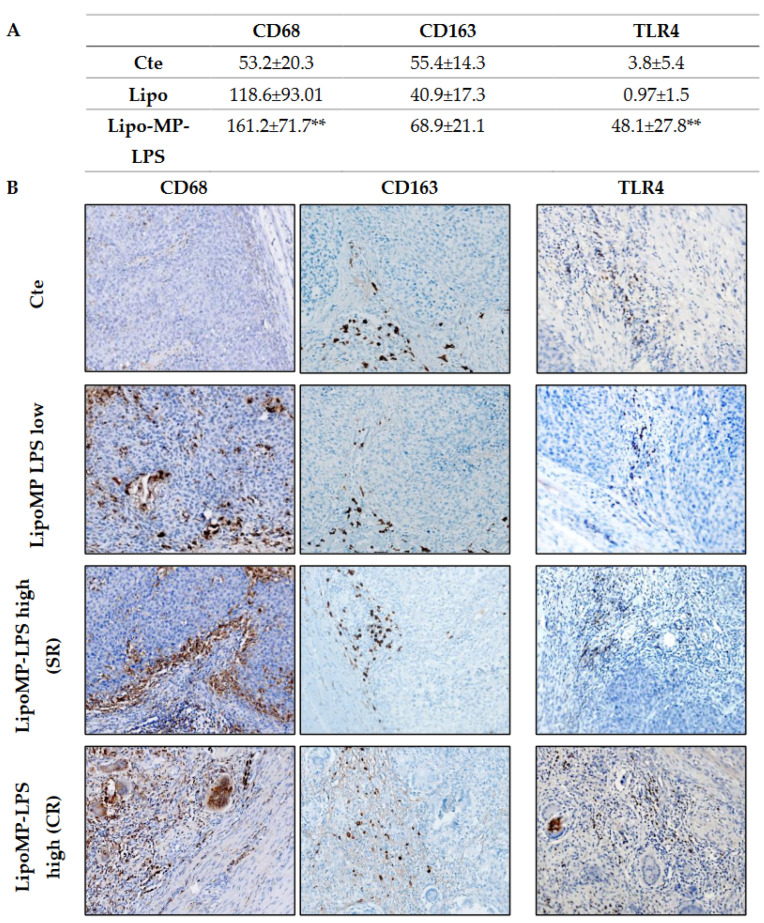
Lipo-MP-LPS modify TAMs infiltrate in OsA’s immune environment. (**A**) Mean density of CD68+, CD163+ and TLR4+ macrophages/FOV according to treatments. Total CD68+ macrophages were significantly increased in LipoMP-LPS High, but not in LipoMP-LPS Low group (both in CR and SR; n = 7) compared with control group (saline, n = 7) while immunosuppressive CD163+ macrophages were only slightly increased. Density of TLR4+ macrophages was also significantly increased in LipoMP-LPS High groups. ** *p* < 0.005; (**B**) Representative staining of CD68 total macrophages, CD163 M2 macrophages and TLR4+ macrophages. Magnification ×200.

**Figure 5 cancers-15-04635-f005:**
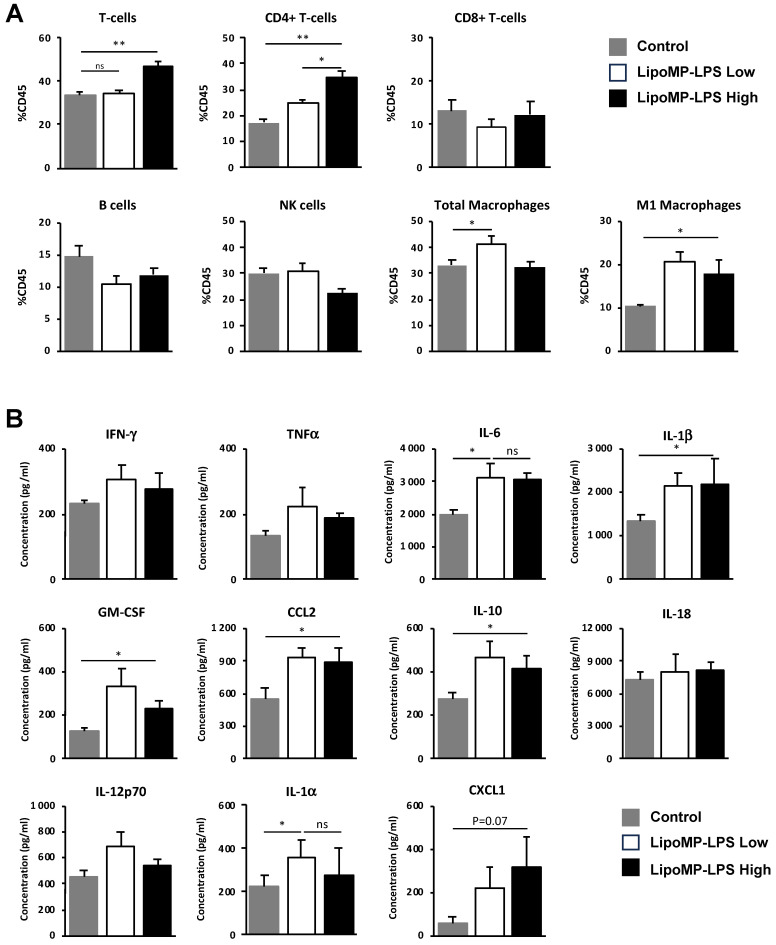
LipoMP-LPS modifies immune cell populations and cytokine environment in tumors at the onset of tumor response. (**A**) Flow cytometry analyses of immune cell populations in the tumors of animal treated with high or low doses of LipoMP-LPS or PBS as control. Data are presented as mean ratio ± SEM of positive cells in each group. ns: not significant * *p* < 0.05, ** *p* < 0.005. High dose of LipoMP-LPS increases tumor infiltration by CD4 T-cells and increases the levels of M1 macrophages but not total macrophages into the tumors; (**B**) Cytokine expression profile evaluated by LegendplexTM in the same tumors. Data are mean ± SEM of the concentration (pg/mL) of each cytokine. High and low doses of LipoMP-LPS induce a moderate proinflammatory environment in tumors, characterized by a slight increase in levels of IL-6, IL-1b, IL-1a, IL-10 and GM-CSF and of the chemokines CCL2 and CXCL1 in tumors compared with control groups. ns: not significant * *p* < 0.05.

**Figure 6 cancers-15-04635-f006:**
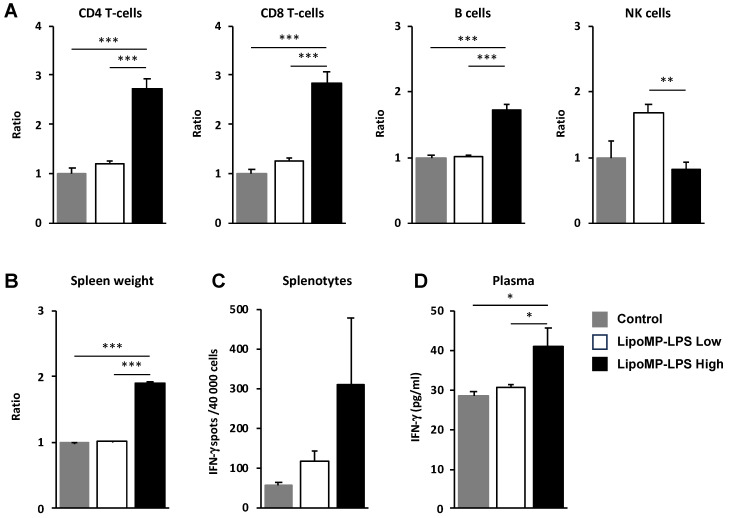
LipoMP-LPS induced systemic adaptive immune responses. (**A**) Flow cytometry analysis of the immune cell populations in the spleen of OsA tumor bearing rats injected with high- or low-dose LipoMP-LPS, or with PBS as controls. For each group, the modification in cells density is expressed as fold change in comparison to control group (set up arbitrary at 1). Data are presented as mean ± SEM ratio of mean percentages of positive cells among CD45+ cells, normalized to mean spleen weights. ** *p* < 0.005; *** *p* < 0.001; (**B**) Mean spleen weights in groups. Data are presented as mean ratio ± SEM of spleen weights. *** *p* < 0.001; (**C**) Analysis of IFN-γ secretion by total splenocytes in each groups using ELISpot assays. Data are mean ± SEM of IFN-γ spots (4 × 10^4^ cells/wells in triplicate). IFN-g production (i.e., number of spots per 40,000 cell/well) was increased in LipoMP-LPS treated group compared with control; (**D**) Plasmatic levels of IFN-γ in each group. Data are mean ± SEM of plasma concentrations of IFN-γ evaluated for 4 animals/treatment group. * *p* < 0.05.

**Figure 7 cancers-15-04635-f007:**
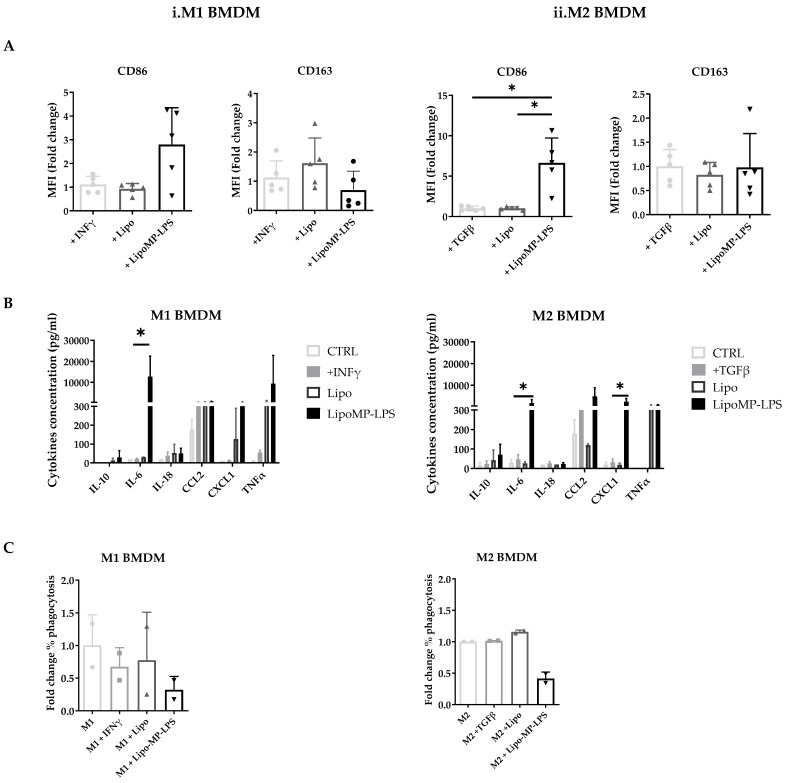
Phenotypes of bone marrow derived macrophages incubated with Lipo-MP-LPS. Rat bone marrow-derived macrophages polarized toward an M1 or M2 phenotype were incubated with LipoMP-LPS, or with PBS or Lipo as controls. (**A**) Flow cytometry analysis of the expression of the surface markers CD86 and CD163 on M1 and M2 polarized macrophages after 24 h of stimulation. Data are presented as mean fold change in MFI/Control ± SD; * *p* < 0.05; (**B**) Cytokines profile of stimulated M1- and M2-polarized macrophages evaluated by LEGENDplex multiplex bead assays. Data are presented as mean concentrations (pg/mL) of each cytokine ± SD. * *p* < 0.05; (**C**) Phagocytosis function of stimulated M1- and M2-polarized macrophages. Data are presented as mean fold change in % phagocytosis/Control ± SD.

## Data Availability

The GSE21257 can be accessed by following the link: https://www.ncbi.nlm.nih.gov/geo/query/acc.cgi?acc=GSE21257 (accessed on 12 July 2019). The GSE33382 can be accessed by following the link: https://www.ncbi.nlm.nih.gov/geo/query/acc.cgi?acc=GSE33382 (accessed on 12 July 2019).

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
