# Peer review of "A TLR4 Agonist Induces Osteosarcoma Regression by Inducing an Antitumor Immune Response and Reprogramming M2 Macrophages to M1 Macrophages"

_cancers, 2023, doi:10.3390/cancers15184635_

Round 1
Reviewer 1 Report
The article is a preclinical evaluation of Lipo-MP-LPS, a TLR4 agonist and in oseteosarcoma model. The paper is well-written and throughly establishes the efficacy of the candidate molecule. There are only a few minor comments:
1. Introduction is too lengthy and can be cut to almost half.
2. The quality of the figures needs to be improved significantly. This version is too pixilated.
3. In the discussion, it would be great if the authors could emphasize on other TLR based immunotherapies in different tumor models and how Lipo-MP-LPS is similar or different.
4. Please quantify IHC data where ever applicable.
Author Response
Dear Sir, Madam,
Thank you for the time you spent reading our manuscript and for the insightful comments you provided. As you suggested in point 1, we have shortened the introduction and improved the quality of the figures (point 2).
As suggested by the reviewer in point 3, we discussed the advantages of TLR4 agonist over other TLR. The recent publication of a study dealing with the use of a TLR9 agonist in an osteosarcoma model (PMID: 37365634) enabled us to make a comparison between these 2 agents used in models of the same tumor. We preferred this approach to comparing the effect of Lipo-MP-LPS with that of other TLR agonists in other tumor histotypes. We briefly discussed the advantage of TLR4 agonists over other TLRs agonists (line 640).
Concerning point 4 mentioned by the reviewer and IHC quantification, the markers of the different immune populations had been quantified as indicated in the previous version of the manuscript (counting carried out for each tumor on 5 non-consecutive FOVs and counting indicated in corresponding tables (cf figures 3 and 4). We have added quantification of cleaved caspase 3 in the organotypic culture model (line 533). OF note, quantification of necrotic areas, Ki67+ cells and treatment-induced remodelling in tumors were viewed and quantified by a sarcoma pathologist (Marie Karanian, added in material sections line 315) using the same method as for quantification of necrotic zones and Ki67+ cells in human sarcomas (assessment of percentages of necrotic zones and percentage of KI67+ cells on the whole tumor slices).
We hope that the modifications we have made to the manuscript and the explanations provided are in line with your expectations and provide sufficient information to improve the manuscript and enable its publication. Thank you for your interest in our work.
Respectfully yours,
Aurélie DUTOUR PhD
Reviewer 2 Report
Overall, this manuscript is of interest. However, the quality of the figures must be improved, and flow cytometry gating and analysis is missing. In addition, some methods (monocytes purification from bone marrow) are not adequately described.
1) The quality of the figure is very low (in particular Figure 2). Figures should be reloaded with higher definition.
2) Figure 5 and 6: representative flow cytometry analysis should be provided for each immune cell population, including the gating.
3) Figure 6: analysis of myeloid cell population in the spleen of Osa tumor-bearing rats is missing. In particular, the role of Lipo-MP-LPS on Myeloid-Derived-Suppressor Cells (MDSC) should be investigated by the authors.
4) Line 287: How monocytes were isolated from bone marrow?
5) Line 290-293 (Macrophages polarization): Authors should explain the M1 and M2 polarization protocol. Why TGFb was used to induce M2-like macrophages? Please, references should be added. Did the authors consider to use IL-4+ IL-13 to induce M2-like macrophages? Did the authors test the impact of Lipo-MP-LPS on Tumor-conditioned macrophages in vitro?
6) References on TME in OS should be updated (for instance, PMID: 32717819 should be included)
Minor editing of English language is required
Author Response
Dear Sir, Madam,
We thank you for the time you spent reading our manuscript and appreciate the very useful comments you provided to improve this manuscript.
- As suggested in point 1, figures were reloaded with higher definition.
- to clarify figures 5 and 6, and as noted in point 2 by the reviewer, the gating strategy used to identify the different immune populations was added as supplementary figure (Fig S1, line 231).
The reviewer notes that we have not presented analyses of myeloid populations in the spleen (point 3). Indeed, in previous experiments under the same conditions, we did not observe any major changes in spleen myeloid populations between the different treatment groups. Thus, we decided to concentrate the analysis of other immune populations present in this organ.
The role of Lipo-MP-LPs on MDSCs is highly relevant and deserves to be studied further. The study presented here was carried out in a rat model, and the tools for distinguishing the different immune populations in rats are less numerous than the one available for murine models, so given these technical limitations we were not able to analyze this point. However, the point raised by the reviewer is important, and we will be able to investigate this in relevant murine models.
TLR4-mediated immunosuppression is linked to the Myd88 signaling pathway. However, the cytokine profile observed in response to Lipo-MP-LPS indicates that this TLR4 agonist directs more towards activation of the TRIF pathway, suggesting that Lipo-MP-LPS has a lesser immunosuppressive effect than other TLR4 agonists.
- In accordance with points 4 and 5, in the material and methods section we have completed the paragraph on monocyte isolation (line 252) and added details on macrophage polarization (references of techniques used were added (line 262)).
To answer the question regarding the use of TGFβ versus the combination of IL-4 + IL-13 to induce M2-like macrophages: this choice was made after preliminary tests carried out to identify the most relevant cytokines (or cocktail of cytokines) which would induce the expression of M2 markers, the production of immunosuppressive cytokines and play a role in phagocytosis potential. TGFβ polarizes macrophages toward M2c phenotype, while the IL-4+IL-13 combination polarizes them toward M2a phenotype. M2c macrophages express CD163, produce immunosuppressive cytokines, and phagocytes a combination of markers that seemed most relevant in our case. This is why we chose to use TGFβ to induce M2-like macrophages.
The last question raised in point 5 concern the evaluation of Lipo-MP-LPS directly on tumor-conditioned macrophages in vitro. We did not carry out this assay but it would be interesting to realize this experiment. In our in vivo study, when tumors begin to respond to Lipo-MP-LPS, we observed an increase in M1-like macrophages in the tumor environment without an increase in total macrophages (Figure 5) and from here conducted the polarization assay in rat macrophages and showed the ability of Lipo-MP-LPS to reorient the M2 profile of macrophages to M1. A similar study was reported by Chettab et al (PMID: 37223101). The authors showed that in vitro Lipo-MP-LPS can induced a polarization reversion of human M2 macrophages toward M1 phenotype.
- We updated the references related to OsA’s TME (point 6)
We hope that the modifications we have made to the manuscript and the explanations provided provide sufficient information to improve the manuscript and enable its publication. Thank you for your interest in our work.
Respectfully yours,
Aurélie DUTOUR PhD

Round 2
Reviewer 2 Report
The new version of this manuscript is sufficiently improved.
Minor revision of English editing is required